# Residual Networks Behave Like Ensembles of Relatively Shallow Networks

**Andreas Veit**    **Michael Wilber**    **Serge Belongie**
Department of Computer Science & Cornell Tech
Cornell University
{av443, mjw285, sjb344}@cornell.edu

## Abstract

In this work we propose a novel interpretation of residual networks showing that they can be seen as a collection of many paths of differing length. Moreover, residual networks seem to enable very deep networks by leveraging only the short paths during training. To support this observation, we rewrite residual networks as an explicit collection of paths. Unlike traditional models, paths through residual networks vary in length. Further, a lesion study reveals that these paths show ensemble-like behavior in the sense that they do not strongly depend on each other. Finally, and most surprising, most paths are shorter than one might expect, and only the short paths are needed during training, as longer paths do not contribute any gradient. For example, most of the gradient in a residual network with 110 layers comes from paths that are only 10-34 layers deep. Our results reveal one of the key characteristics that seem to enable the training of very deep networks: Residual networks avoid the vanishing gradient problem by introducing short paths which can carry gradient throughout the extent of very deep networks.

## 1   Introduction

Most modern computer vision systems follow a familiar architecture, processing inputs from low-level features up to task specific high-level features. Recently proposed residual networks [5, 6] challenge this conventional view in three ways. First, they introduce *identity skip-connections* that bypass residual layers, allowing data to flow from any layers directly to any subsequent layers. This is in stark contrast to the traditional strictly sequential pipeline. Second, skip connections give rise to networks that are *two orders of magnitude deeper* than previous models, with as many as 1202 layers. This is contrary to architectures like AlexNet [13] and even biological systems [17] that can capture complex concepts within half a dozen layers.[1] Third, in initial experiments, we observe that *removing single layers from residual networks at test time does not noticeably affect their performance*. This is surprising because removing a layer from a traditional architecture such as VGG [18] leads to a dramatic loss in performance.

In this work we investigate the impact of these differences. To address the influence of identity skip-connections, we introduce the *unraveled view*. This novel representation shows residual networks can be viewed as a collection of many paths instead of a single deep network. Further, the perceived resilience of residual networks raises the question whether the paths are dependent on each other or whether they exhibit a degree of redundancy. To find out, we perform a *lesion study*. The results show ensemble-like behavior in the sense that removing paths from residual networks by deleting layers or corrupting paths by reordering layers only has a modest and smooth impact on performance. Finally, we investigate the depth of residual networks. Unlike traditional models, paths through residual networks vary in length. The distribution of path lengths follows a binomial distribution, meaning

that the majority of paths in a network with 110 layers are only about 55 layers deep. Moreover, we show most gradient during training comes from paths that are even shorter, i.e., 10-34 layers deep.

This reveals a tension. On the one hand, residual network performance improves with adding more and more layers [6]. However, on the other hand, residual networks can be seen as collections of many paths and the only effective paths are relatively shallow. Our results could provide a first explanation: residual networks do not resolve the vanishing gradient problem by preserving gradient flow throughout the entire depth of the network. Rather, they enable very deep networks by shortening the effective paths. For now, short paths still seem necessary to train very deep networks.

In this paper we make the following contributions:

- We introduce the unraveled view, which illustrates that residual networks can be viewed as a collection of many paths, instead of a single ultra-deep network.
- We perform a lesion study to show that these paths do not strongly depend on each other, even though they are trained jointly. Moreover, they exhibit ensemble-like behavior in the sense that their performance smoothly correlates with the number of valid paths.
- We investigate the gradient flow through residual networks, revealing that only the short paths contribute gradient during training. Deep paths are not required during training.

## 2   Related Work

**The sequential and hierarchical computer vision pipeline** Visual processing has long been understood to follow a hierarchical process from the analysis of simple to complex features. This formalism is based on the discovery of the receptive field [10], which characterizes the visual system as a hierarchical and feedforward system. Neurons in early visual areas have small receptive fields and are sensitive to basic visual features, e.g., edges and bars. Neurons in deeper layers of the hierarchy capture basic shapes, and even deeper neurons respond to full objects. This organization has been widely adopted in the computer vision and machine learning literature, from early neural networks such as the Neocognitron [4] and the traditional hand-crafted feature pipeline of Malik and Perona [15] to convolutional neural networks [13, 14]. The recent strong results of very deep neural networks [18, 20] led to the general perception that it is the depth of neural networks that govern their expressive power and performance. In this work, we show that residual networks do not necessarily follow this tradition.

**Residual networks** [5, 6] are neural networks in which each layer consists of a residual module $f_i$ and a skip connection[2] bypassing $f_i$. Since layers in residual networks can comprise multiple convolutional layers, we refer to them as residual blocks in the remainder of this paper. For clarity of notation, we omit the initial pre-processing and final classification steps. With $y_{i-1}$ as is input, the output of the $i$th block is recursively defined as

$$y_i \equiv f_i(y_{i-1}) + y_{i-1}, \tag{1}$$

where $f_i(x)$ is some sequence of convolutions, batch normalization [11], and Rectified Linear Units (ReLU) as nonlinearities. Figure 1 (a) shows a schematic view of this architecture. In the most recent formulation of residual networks [6], $f_i(x)$ is defined by

$$f_i(x) \equiv W_i \cdot \sigma \left( B \left( W_i' \cdot \sigma \left( B \left( x \right) \right) \right) \right), \tag{2}$$

where $W_i$ and $W_i'$ are weight matrices, $\cdot$ denotes convolution, $B(x)$ is batch normalization and $\sigma(x) \equiv \max(x, 0)$. Other formulations are typically composed of the same operations, but may differ in their order.

The idea of branching paths in neural networks is not new. For example, in the regime of convolutional neural networks, models based on inception modules [20] were among the first to arrange layers in blocks with parallel paths rather than a strict sequential order. We choose residual networks for this study because of their simple design principle.

**Highway networks** Residual networks can be viewed as a special case of highway networks [19]. The output of each layer of a highway network is defined as

$$y_{i+1} \equiv f_{i+1}(y_i) \cdot t_{i+1}(y_i) + y_i \cdot (1 - t_{i+1}(y_i)) \tag{3}$$

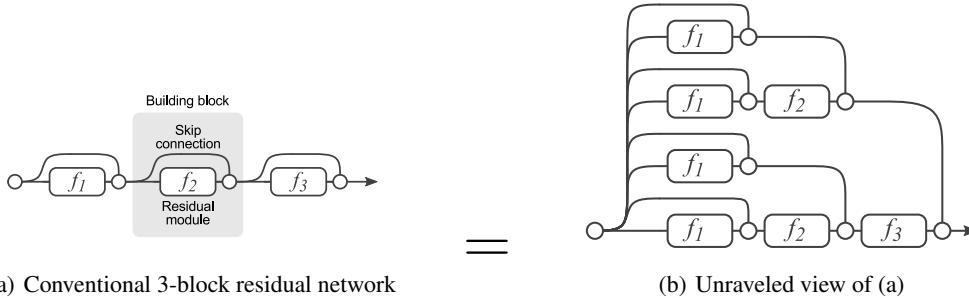

(a) Conventional 3-block residual network        (b) Unraveled view of (a)

Figure 1: Residual Networks are conventionally shown as (a), which is a natural representation of Equation (1). When we expand this formulation to Equation (6), we obtain an *unraveled view* of a 3-block residual network (b). Circular nodes represent additions. From this view, it is apparent that residual networks have $O(2^n)$ implicit paths connecting input and output and that adding a block doubles the number of paths.

This follows the same structure as Equation (1). Highway networks also contain residual modules and skip connections that bypass them. However, the output of each path is attenuated by a gating function $t$, which has learned parameters and is dependent on its input. Highway networks are equivalent to residual networks when $t_i(\cdot) = 0.5$, in which case data flows equally through both paths. Given an omnipotent solver, highway networks could learn whether each residual module should affect the data. This introduces more parameters and more complexity.

**Investigating neural networks** Several investigative studies seek to better understand convolutional neural networks. For example, Zeiler and Fergus [23] visualize convolutional filters to unveil the concepts learned by individual neurons. Further, Szegedy et al. [21] investigate the function learned by neural networks and how small changes in the input called adversarial examples can lead to large changes in the output. Within this stream of research, the closest study to our work is from Yosinski et al. [22], which performs lesion studies on AlexNet. They discover that early layers exhibit little co-adaptation and later layers have more co-adaptation. These papers, along with ours, have the common thread of exploring specific aspects of neural network performance. In our study, we focus our investigation on structural properties of neural networks.

**Ensembling** Since the early days of neural networks, researchers have used simple ensembling techniques to improve performance. Though boosting has been used in the past [16], one simple approach is to arrange a committee [3] of neural networks in a simple voting scheme, where the final output predictions are averaged. Top performers in several competitions use this technique almost as an afterthought [6, 13, 18]. Generally, one key characteristic of ensembles is their smooth performance with respect to the number of members. In particular, the performance increase from additional ensemble members gets smaller with increasing ensemble size. Even though they are not strict ensembles, we show that residual networks behave similarly.

**Dropout** Hinton et al. [7] show that dropping out individual neurons during training leads to a network that is equivalent to averaging over an ensemble of exponentially many networks. Similar in spirit, stochastic depth [9] trains an ensemble of networks by dropping out entire layers during training. In this work, we show that one does not need a special training strategy such as stochastic depth to drop out layers. Entire layers can be removed from plain residual networks without impacting performance, indicating that they do not strongly depend on each other.

## 3 The unraveled view of residual networks

To better understand residual networks, we introduce a formulation that makes it easier to reason about their recursive nature. Consider a residual network with three building blocks from input $y_0$ to output $y_3$. Equation (1) gives a recursive definition of residual networks. The output of each stage is based on the combination of two subterms. We can make the shared structure of the residual network apparent by unrolling the recursion into an exponential number of nested terms, expanding one layer

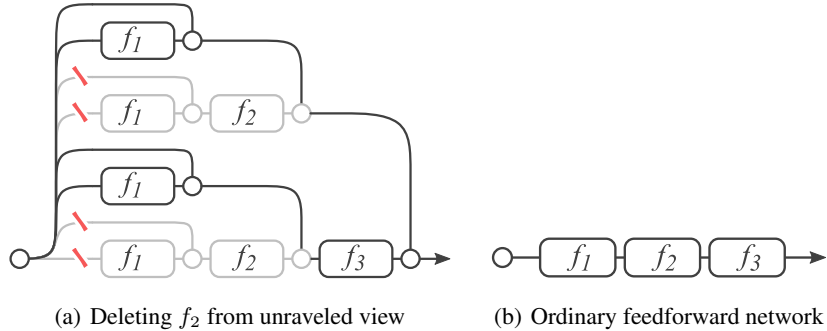

| (a) Deleting $f_2$ from unraveled view | (b) Ordinary feedforward network |

Figure 2: Deleting a layer in residual networks at test time (a) is equivalent to zeroing half of the paths. In ordinary feed-forward networks (b) such as VGG or AlexNet, deleting individual layers alters the only viable path from input to output.

at each substitution step:

$$y_3 = y_2 + f_3(y_2) \tag{4}$$
$$= [y_1 + f_2(y_1)] \ + \ f_3(y_1 + f_2(y_1)) \tag{5}$$
$$= \big[y_0 + f_1(y_0) + f_2(y_0 + f_1(y_0))\big] \ + \ f_3\big(y_0 + f_1(y_0) + f_2(y_0 + f_1(y_0))\big) \tag{6}$$

We illustrate this expression tree graphically in Figure 1 (b). With subscripts in the function modules indicating weight sharing, this graph is equivalent to the original formulation of residual networks. The graph makes clear that data flows along many *paths* from input to output. Each path is a unique configuration of which residual module to enter and which to skip. Conceivably, each unique path through the network can be indexed by a binary code $b \in \{0, 1\}^n$ where $b_i = 1$ iff the input flows through residual module $f_i$ and 0 if $f_i$ is skipped. It follows that residual networks have $2^n$ paths connecting input to output layers.

In the classical visual hierarchy, each layer of processing depends *only* on the output of the previous layer. Residual networks cannot strictly follow this pattern because of their inherent structure. Each module $f_i(\cdot)$ in the residual network is fed data from a mixture of $2^{i-1}$ different distributions generated from every possible configuration of the previous $i - 1$ residual modules.

Compare this to a strictly sequential network such as VGG or AlexNet, depicted conceptually in Figure 2 (b). In these networks, input always flows from the first layer straight through to the last in a single path. Written out, the output of a three-layer feed-forward network is

$$y_3^{FF} = f_3^{FF}(f_2^{FF}(f_1^{FF}(y_0))) \tag{7}$$

where $f_i^{FF}(x)$ is typically a convolution followed by batch normalization and ReLU. In these networks, each $f_i^{FF}$ is only fed data from a single path configuration, the output of $f_{i-1}^{FF}(\cdot)$.

It is worthwhile to note that ordinary feed-forward neural networks can also be "unraveled" using the above thought process at the level of individual neurons rather than layers. This renders the network as a collection of different paths, where each path is a unique configuration of neurons from each layer connecting input to output. Thus, all paths through ordinary neural networks are of the same length. However, paths in residual networks have varying length. Further, each path in a residual network goes through a different subset of layers.

Based on these observations, we formulate the following questions and address them in our experiments below. Are the paths in residual networks dependent on each other or do they exhibit a degree of redundancy? If the paths do not strongly depend on each other, do they behave like an ensemble? Do paths of varying lengths impact the network differently?

## 4 Lesion study

In this section, we use three lesion studies to show that paths in residual networks do not strongly depend on each other and that they behave like an ensemble. All experiments are performed at test

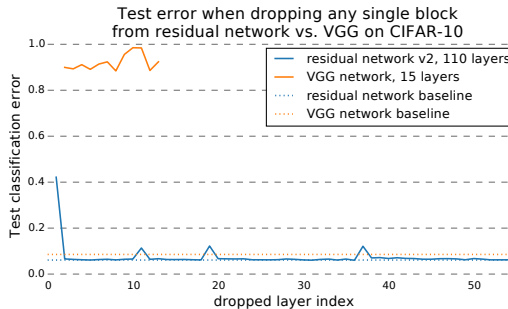
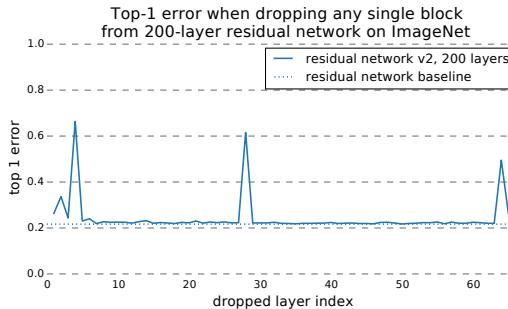

Figure 3: Deleting individual layers from VGG and a residual network on CIFAR-10. VGG performance drops to random chance when any one of its layers is deleted, but deleting individual modules from residual networks has a minimal impact on performance. Removing downsampling modules has a slightly higher impact.

Figure 4: Results when dropping individual blocks from residual networks trained on ImageNet are similar to CIFAR results. However, downsampling layers tend to have more impact on ImageNet.

time on CIFAR-10 [12]. Experiments on ImageNet [2] show comparable results. We train residual networks with the standard training strategy, dataset augmentation, and learning rate policy, [6]. For our CIFAR-10 experiments, we train a 110-layer (54-module) residual network with modules of the "pre-activation" type which contain batch normalization as first step. For ImageNet we use 200 layers (66 modules). It is important to note that we did not use any special training strategy to adapt the network. In particular, we did *not* use any perturbations such as stochastic depth during training.

## 4.1 Experiment: Deleting individual layers from neural networks at test time

As a motivating experiment, we will show that not all transformations within a residual network are necessary by deleting individual modules from the neural network after it has been fully trained. To do so, we remove the residual module from a single building block, leaving the skip connection (or downsampling projection, if any) untouched. That is, we change $y_i = y_{i-1} + f_i(y_{i-1})$ to $y_i' = y_{i-1}$. We can measure the importance of each building block by varying which residual module we remove. To compare to conventional convolutional neural networks, we train a VGG network with 15 layers, setting the number of channels to 128 for all layers to allow the removal of any layer.

It is unclear whether any neural network can withstand such a drastic change to the model structure. We expect them to break because dropping any layer drastically changes the input distribution of all subsequent layers.

The results are shown in Figure 3. As expected, deleting any layer in VGG reduces performance to chance levels. Surprisingly, **this is not the case for residual networks**. Removing downsampling blocks does have a modest impact on performance (peaks in Figure 3 correspond to downsampling building blocks), but no other block removal lead to a noticeable change. This result shows that to some extent, the structure of a residual network can be changed at runtime without affecting performance. Experiments on ImageNet show comparable results, as seen in Figure 4.

Why are residual networks resilient to dropping layers but VGG is not? Expressing residual networks in the unraveled view provides a first insight. It shows that residual networks can be seen as a collection of many paths. As illustrated in Figure 2 (a), when a layer is removed, the number of paths is reduced from $2^n$ to $2^{n-1}$, leaving half the number of paths valid. VGG only contains a single usable path from input to output. Thus, when a single layer is removed, the only viable path is corrupted. *This result suggests that paths in a residual network do not strongly depend on each other although they are trained jointly.*

## 4.2 Experiment: Deleting many modules from residual networks at test-time

Having shown that paths do not strongly depend on each other, we investigate whether the collection of paths shows ensemble-like behavior. One key characteristic of ensembles is that their performance

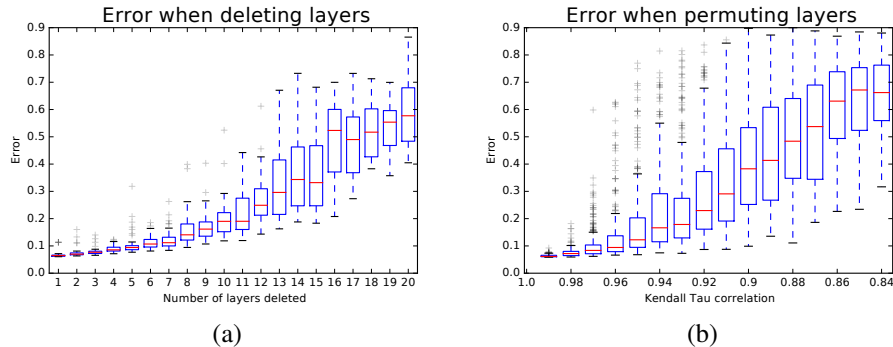

Figure 5: (a) Error increases smoothly when randomly deleting several modules from a residual network. (b) Error also increases smoothly when re-ordering a residual network by shuffling building blocks. The degree of reordering is measured by the Kendall Tau correlation coefficient. These results are similar to what one would expect from ensembles.

depends smoothly on the number of members. If the collection of paths were to behave like an ensemble, we would expect test-time performance of residual networks to smoothly correlate with the *number of valid paths*. This is indeed what we observe: deleting increasing numbers of residual modules, increases error smoothly, Figure 5 (a). This implies residual networks behave like ensembles.

When deleting $k$ residual modules from a network originally of length $n$, the number of valid paths decreases to $O(2^{n-k})$. For example, the original network started with 54 building blocks, so deleting 10 blocks leaves $2^{44}$ paths. Though the collection is now a factor of roughly $10^{-6}$ of its original size, there are still many valid paths and error remains around 0.2.

### 4.3   Experiment: Reordering modules in residual networks at test-time

Our previous experiments were only about dropping layers, which have the effect of removing paths from the network. In this experiment, we consider changing the structure of the network by *re-ordering* the building blocks. This has the effect of removing some paths and inserting new paths that have never been seen by the network during training. In particular, it moves high-level transformations before low-level transformations.

To re-order the network, we swap $k$ randomly sampled pairs of building blocks with compatible dimensionality, ignoring modules that perform downsampling. We graph error with respect to the Kendall Tau rank correlation coefficient which measures the amount of corruption. The results are shown in Figure 5 (b). As corruption increases, the error smoothly increases as well. This result is surprising because it suggests that residual networks can be reconfigured to some extent at runtime.

## 5   The importance of short paths in residual networks

Now that we have seen that there are many paths through residual networks and that they do not necessarily depend on each other, we investigate their characteristics.

**Distribution of path lengths**   Not all paths through residual networks are of the same length. For example, there is precisely one path that goes through all modules and $n$ paths that go only through a single module. From this reasoning, the distribution of all possible path lengths through a residual network follows a Binomial distribution. Thus, we know that the path lengths are closely centered around the mean of $n/2$. Figure 6 (a) shows the path length distribution for a residual network with 54 modules; more than 95% of paths go through 19 to 35 modules.

**Vanishing gradients in residual networks**   Generally, data flows along all paths in residual networks. However, not all paths carry the same amount of gradient. In particular, the length of the paths through the network affects the gradient magnitude during backpropagation [1, 8]. To empirically investigate the effect of vanishing gradients on residual networks we perform the following experiment. Starting from a trained network with 54 blocks, we sample *individual paths* of a certain length and measure the norm of the gradient that arrives at the input. To sample a path of length $k$, we first feed a batch forward through the whole network. During the backward pass, we randomly sample $k$ residual

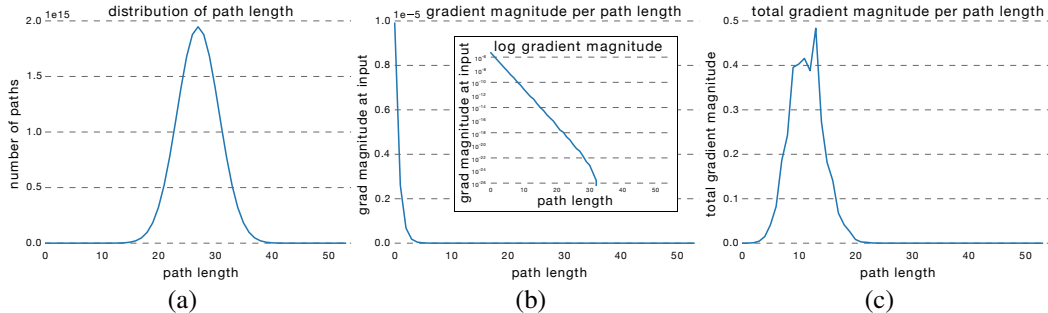

Figure 6: How much gradient do the paths of different lengths contribute in a residual network? To find out, we first show the distribution of all possible path lengths (a). This follows a Binomial distribution. Second, we record how much gradient is induced on the first layer of the network through paths of varying length (b), which appears to decay roughly exponentially with the number of modules the gradient passes through. Finally, we can multiply these two functions (c) to show how much gradient comes from all paths of a certain length. Though there are many paths of medium length, paths longer than $\sim$20 modules are generally too long to contribute noticeable gradient during training. This suggests that the effective paths in residual networks are relatively shallow.

blocks. For those $k$ blocks, we only propagate through the residual module; for the remaining $n - k$ blocks, we only propagate through the skip connection. Thus, we only measure gradients that flow through the single path of length $k$. We sample 1,000 measurements for each length $k$ using random batches from the training set. The results show that the gradient magnitude of a path decreases exponentially with the number of modules it went through in the backward pass, Figure 6 (b).

**The effective paths in residual networks are relatively shallow** Finally, we can use these results to deduce whether shorter or longer paths contribute most of the gradient during training. To find the total gradient magnitude contributed by paths of each length, we multiply the frequency of each path length with the expected gradient magnitude. The result is shown in Figure 6 (c). Surprisingly, almost all of the gradient updates during training come from paths between 5 and 17 modules long. These are the *effective paths*, even though they constitute only $0.45\%$ of all paths through this network. Moreover, in comparison to the total length of the network, the effective paths are *relatively shallow*.

To validate this result, we retrain a residual network from scratch that only sees the effective paths during training. This ensures that no long path is ever used. If the retrained model is able to perform competitively compared to training the full network, we know that long paths in residual networks are not needed during training. We achieve this by only training a subset of the modules during each mini batch. In particular, we choose the number of modules such that the distribution of paths during training aligns with the distribution of the effective paths in the whole network. For the network with 54 modules, this means we sample exactly 23 modules during each training batch. Then, the path lengths during training are centered around 11.5 modules, well aligned with the effective paths. In our experiment, the network trained only with the effective paths achieves a 5.96% error rate, whereas the full model achieves a 6.10% error rate. There is no statistically significant difference. This demonstrates that indeed only the effective paths are needed.

## 6 Discussion

**Removing residual modules mostly removes long paths** Deleting a module from a residual network mainly removes the long paths through the network. In particular, when deleting $d$ residual modules from a network of length $n$, the fraction of paths remaining per path length $x$ is given by

$$\text{fraction of remaining paths of length } x = \frac{\binom{n-d}{x}}{\binom{n}{x}} \tag{8}$$

Figure 7 illustrates the fraction of remaining paths after deleting 1, 10 and 20 modules from a 54 module network. It becomes apparent that the deletion of residual modules mostly affects the long paths. Even after deleting 10 residual modules, many of the *effective paths* between 5 and 17 modules long are still valid. Since mainly the effective paths are important for performance, this result is in line with the experiment shown in Figure 5 (a). Performance only drops slightly up to the removal of 10 residual modules, however, for the removal of 20 modules, we observe a severe drop in performance.

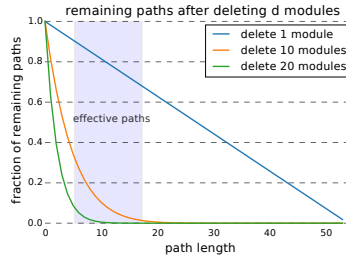
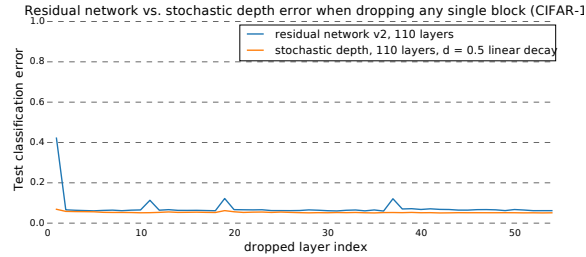

Figure 7: Fraction of paths remaining after deleting individual layers. Deleting layers mostly affects long paths through the networks.

Figure 8: Impact of stochastic depth on resilience to layer deletion. Training with stochastic depth only improves resilience slightly, indicating that plain residual networks already don't depend on individual layers. Compare to Fig. 3.

**Connection to highway networks** In highway networks, $t_i(\cdot)$ multiplexes data flow through the residual and skip connections and $t_i(\cdot) = 0.5$ means both paths are used equally. For highway networks in the wild, [19] observe empirically that the gates commonly deviate from $t_i(\cdot) = 0.5$. In particular, they tend to be biased toward sending data through the skip connection; in other words, the network learns to use short paths. Similar to our results, it reinforces the importance of short paths.

**Effect of stochastic depth training procedure** Recently, an alternative training procedure for residual networks has been proposed, referred to as stochastic depth [9]. In that approach a random subset of the residual modules is selected for each mini-batch during training. The forward and backward pass is only performed on those modules. Stochastic depth does not affect the number of paths in the network because all paths are available at test time. However, it changes the distribution of paths seen during training. In particular, mainly short paths are seen. Further, by selecting a different subset of short paths in each mini-batch, it encourages the paths to produce good results independently.

Does this training procedure significantly reduce the dependence between paths? We repeat the experiment of deleting individual modules for a residual network trained using stochastic depth. The result is shown in Figure 8. Training with stochastic depth improves resilience slightly; only the dependence on the downsampling layers seems to be reduced. By now, this is not surprising: we know that plain residual networks already don't depend on individual layers.

# 7 Conclusion

What is the reason behind residual networks' increased performance? In the most recent iteration of residual networks, He et al. [6] provide one hypothesis: "We obtain these results via a simple but essential concept—going deeper." While it is true that they are deeper than previous approaches, we present a complementary explanation. First, our unraveled view reveals that residual networks can be viewed as a collection of many paths, instead of a single ultra deep network. Second, we perform lesion studies to show that, although these paths are trained jointly, they do not strongly depend on each other. Moreover, they exhibit ensemble-like behavior in the sense that their performance smoothly correlates with the number of valid paths. Finally, we show that the paths through the network that contribute gradient during training are shorter than expected. In fact, deep paths are not required during training as they do not contribute any gradient. Thus, residual networks do not resolve the vanishing gradient problem by preserving gradient flow throughout the entire depth of the network. This insight reveals that depth is still an open research question. These promising observations provide a new lens through which to examine neural networks.

# Acknowledgements

We would like to thank Sam Kwak and Theofanis Karaletsos for insightful feedback. We also thank the reviewers of NIPS 2016 for their very constructive and helpful feedback and for suggesting the paper title. This work is partly funded by AOL through the Connected Experiences Laboratory (Author 1), an NSF Graduate Research Fellowship award (NSF DGE-1144153, Author 2), and a Google Focused Research award (Author 3).

## Footnotes

[1]Making the common assumption that a layer in a neural network corresponds to a cortical area.

[2]We only consider identity skip connections, but this framework readily generalizes to more complex projection skip connections when downsampling is required.

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
