[Reviews · NeurIPS 2016]

Reviewer 1

Summary

This paper investigates the inner workings of the residual network architecture recently proposed by Kaiming He et al. The point of departure for the investigation is the claim that a residual network is an "exponential ensemble of relatively shallow networks." After making some mathematical arguments to this point in Section 3, they run several experiments in subsequent sections that show that Resnet behaves like an ensemble in several respects.

Qualitative Assessment

I think the experiments in this paper are very interesting and reveal interesting properties of Resnet. However, I don't think the claim that ResNets are "exponential ensembles" has been made adequetely. If the claim were weakened to "behave like exponential ensembles", this would seem to be reasonably well supported by the experiments, and it's also an interesting result. But I think the question of why this happens is still open. One thing that stands out is that no definition of "ensemble" is ever given, not even a rough definition. My guess is that you have in mind a bagging-style ensemble, where you have a simple combination of a set of models, any subset of which would also be a reasonable ensemble. When approaching the ensemble question from first principles in section 3, you present the "unraveled view" (e.g. equation 6). From this expression for a Resnet, it's clear that we do not have a simple combination of exponentially many models. At best, we can interpret (6) as combining 4 models. I did note, however, that the gradient calculations DO have the form of a sum over exponentially many "paths". Perhaps that's what you have in mind... A similar claim of an "exponential ensemble" is made about dropout in [7], by Hinton et al. There the connection was that for every minibatch, the randomly chosen set of nodes constitute a network that is trained with a single minibatch step. For dropout, it is clear what the exponentially many networks are, but it is not immediately clear that the final prediction can be interpreted as an ensemble of these networks. This link is made (at least in a particular case) by their claim that "in networks with a single hidden layer of N units and a “softmax” output layer for computing the probabilities of the class labels, using the mean network is exactly equivalent to taking the geometric mean of the probability distributions over labels predicted by all 2^N possible networks." Perhaps you could leverage this theorem or something similar to tighten the connection between the resnet and a combination of exponentially many "paths" in resnet. I think that the experiments are interesting and meaningful, but the interpretation and explanation fall short of justifying the title of the paper. Still, I think the results are interesting enough that it would be nice to release the results. I call this paper a borderline and see what others have to say.

Confidence in this Review

2-Confident (read it all; understood it all reasonably well)


Reviewer 2

Summary

The paper studies deep residual networks and puts forward a hypothesis that they can be seen as an ensemble of shallow nets (the authors introduce the notion of multiplicity to denote the size of this implicit ensemble). This is visually demonstrated by unrolling the residual connections ("unraveled view") and showing the net does resemble an ensemble of an exponential number of subnets. To further support this view, several experiments are presented, e.g. it is demonstrated that the network accuracy gradually reduces as the number of layers is decreased at test time, or that most of the weights gradient is contributed by shorter paths within a network.

Qualitative Assessment

Positives: 1) This is an experimental paper, addressing a relevant problem of studying the behaviour of deep residual networks. 2) The ensemble hypothesis is interesting and could be worth presenting at NIPS. Negatives: 1) There is no algorithmic novelty as such. 2) The conclusions derived from the experiments are rather over-assertive, and the paper would benefit from a less aggressive exposition of the experiments. Basically, all conclusions are tied to the message "ResNets are ensembles of shallow nets, it is wrong to say that they are very deep nets". I don't really see a big contradiction here, as the proposed ensemble view is just another way of interpreting the computation carried out by a net with skip connections. Conclusion: this paper can be borderline-accepted, provided that the authors make an effort to adjust the conclusions.

Confidence in this Review

3-Expert (read the paper in detail, know the area, quite certain of my opinion)


Reviewer 3

Summary

The paper deconstructs the recursive structure residual networks, providing evidence to the similarity of their behavior and ensemble methods and thereby framing residual networks as exponential ensembles. A close look at resiliency in the face of removed modules/layers and paths shows the multiplicity, or implicit ensemble size, by construction of residual networks. The authors also look at how gradients are distributed over paths of different length within the network, empirically showing that the network’s depth does not prevent the vanishing gradient problem but rather the depth affords multiple effective gradient paths of reasonable length that help avoid the problem.

Qualitative Assessment

The paper is well written and performs an adequate background search. The methodology for residual networks and experimental results are presented clearly. This paper serves as an excellent introduction to the power of residual networks as this is an emerging method for deep network construction, one of the most confusing and important aspects of deep learning research today. The description of how residual networks increase gradient pathways and mimic ensembles is convincing and satisfying, although it would be especially interesting if removing layers were frame practically (e.g. transferring deep networks to less computationally compatible devices). Going forward I’d like to see more guidance on how residual modules should be used in general network design and hard proofs on how they increase expression potential. One interesting framing of these networks that was not explicitly touched on would be the potential of the skip connection to act as a regularization method to avoid overfitting. In this context an in depth comparison of regularization / ensemble inducing methods such as stochastic pooling, dropout, and stochastic depth training would be interesting but may be beyond scope here. Page 6 Line 186 Missing .

Confidence in this Review

2-Confident (read it all; understood it all reasonably well)


Reviewer 4

Summary

This paper studies the recently proposed resnets architectures from a new perspective, interpreting them as exponential ensembles. The work interprets every residual block in an “unraveled” view where 2 paths exist - through the non-linearity and the skip connection. Thus, input for the i-th layer passes through 2^(i-1) paths and thus, resnets are exponential ensembles. To further prove this, the work performs the following three experiments: (1) Deleting single layers (2) Deleting multiple modules during test time (3) re-ordering modules From these three experiments, it is empirically shown that resnets behave similar to ensembles of relatively “shallow” networks.

Qualitative Assessment

1. The unraveled view of the resnet module is mathematically accurate only when the residual functions f are linear. For example, (1+f_1)(1+f_2) = 1 + f_1 + f_2 + f_1*f_2 only if f_1 and f_2 are linear. However, most common layers are non-linear (piece-wise linear if activation is ReLU ) and hence, the basis of the paper is not mathematically precise. If non-linearity is assumed there exist only i+1 paths as against 2^(i-1) paths making it a linear ensemble. 2. Gradients to layers can be low in two cases - when the filter is already trained ( typically in the initial layers where gaussian filters are learned) or due to vanishing gradients. The argument in the paper will be more convincing if a plot of abs(gradients) vs. ( train - test loss ) vs. iterations is plotted.

Confidence in this Review

3-Expert (read the paper in detail, know the area, quite certain of my opinion)


Reviewer 5

Summary

This paper addresses a new way to analyze residual networks as ensembles of many shallow networks. Following the point of view, residual networks are decomposed to exponentially many networks that are relatively shallow than the original networks. To support the hypothesis, three lesion studies are introduced with experiments for dropping or reordering some residual modules at test time. Discussions for path lengths and vanishing gradients by seeing the gradient flows of modules are presented to support the hypothesis as well.

Qualitative Assessment

This paper introduces a new way to interpret residual networks. This is perhaps the first attempt to analyze the residual networks in relation to model ensembles. This paper is well written, and the main contribution is clearly presented as well. The hypothesis - residual networks are ensembles by construction is quite well supported by the experiments and discussions. The characteristics of residual networks are well-analyzed by observing gradient flows: 1) the effective length of paths is shown to be much smaller than the original length, and 2) the gradients whose magnitudes is 0 are shown on lengthy paths, and this clearly shows the vanishing gradient is not solved by residual networks. This approach is well proposed to utilize other networks related to residual networks. The additional comments are 1) It may be helpful to describe 'ensemble' in terms of weight sharing at training. Since traditional ensembles of models for learning tasks (e.g. classification) are trained independently without sharing weights and are combined together in the test stage after training. However, in residual networks, all the networks with neurons are trained together by sharing weights in a single training, so the meaning of ensembles is quite different. 2) The third experiment in the lesion studies has not enough ground to support that residual networks are ensembles. 3) As mentioned by the authors in P3, stochastic depth is ''ensembles by training'', but otherwise, the authors contradictively state that stochastic depth does not induce the ensemble in P8. 4) Furthermore, stochastic depth seems to improve the robustness of networks much by model ensembles by training. Thus, subsection - 'Effect of stochastic depth training procedure' does not seem to support the hypothesis. P5, 'Deleting 10 layers' is better to be changed by 'deleting 10 modules'. P6. A punctuation mark is missing. P1 ~ P8. It is better not to use the term - 'module' mixed with the term - 'block'.

Confidence in this Review

2-Confident (read it all; understood it all reasonably well)


Reviewer 6

Summary

The authors give a novel interpretation of deep residual networks, a feed-forward architecture that has received significant attention from the research community recently. They state that residual networks are ensembles of relatively shallow networks. They provide detailed experiments to underpin their interpretation.

Qualitative Assessment

The paper approaches an important and interesting question with a novel methodology. It's an inspiring, thought-provoking read, especially for those of us interested in residual networks. It provokes us into looking at resnets (and deep nets in general) from a different perspective. The single big issue I see with it is that its claims are either not formalized enough, and inasmuch as they are formalized, they are not sufficiently justified by the (admittedly extemely well-thought-out and informative) experiments presented. Below are a couple of examples to make this critique a bit more specific. lines 157-158: Here the authors argue for the ensemble view by invoking the observation that single blocks can be removed from the network without significant performance degradation. I claim that this line of argument is not logically sound. A simpler explanation of the robustness phenomenon is that the magnitude of individual residual outputs is small compared to their summed magnitude. (I've actually conducted experiments on deep residual networks that verify this.) Of course, one could use the word "ensemble" in such a general sense that this alternative explanation (of small individual resnet block magnitudes) is not in fact an alternative explanation, but rather, it supports the ensemble claim. I'd say that in this case the ensemble claim is too vacuous. In any case, it's never formalized enough so that the reader can decide. One other way of highlighting the IMHO problematic line of argument is this: A particularly useless kind of ensemble is the echo chamber, where the individual elements of the ensemble all vote the same way. I believe that the arguments and experiments presented in the paper never successfully exclude the possibility that the observed ensemble is of this kind. A related observation: In principle, adding a constant vector "drift" to the output of a residual block does not hurt the learning capabilities of the system: the following blocks simply must learn to work on a translated version of their original input domain. But -- unless those blocks learn to operate both on the translated and the original domain, an a priori unexpected form of generalization -- the artificial removal of the residual block will damage the system, because it removes the drift as well. The experiments presented in the paper clearly show that this degradation does not actually happen: there's no significant drift. It would be interesting to understand this phenomenon better. Is it related to the use of ReLUs or batchnorm? Or is it something more general? I have a vague general issue with the intuition behind the "unraveled view" of the network. The easiest way to state this is through a crude analogy: We can look at the "unraveled view" of the multilinear formula (1+x_1)(1+x_2)...(1+x_n): a polynomial in standard form, with 2^n tems. It's hard to imagine what the unraveled view adds to the discussion in this specific case. When compared to the 1-multiplicity formula x_1x_2...x_n, there are very few meaningful senses of the word in which our 2^n-multiplicity multilinear formula is exponentially more complex. I'm aware that multiplication and function composition are mixed here. It's an imperfect analogy. Still, I would recommend the authors to defuse similar kinds of objections in advance. The above criticisms should not obscure the fact that I've found the paper thought-provoking, the experiments well-designed, and the potential impact high.

Confidence in this Review

3-Expert (read the paper in detail, know the area, quite certain of my opinion)